# The Expression of Cell Cycle-Related Genes in *USP8*-Mutated Corticotroph Neuroendocrine Pituitary Tumors and Their Possible Role in Cell Cycle-Targeting Treatment

**DOI:** 10.3390/cancers14225594

**Published:** 2022-11-14

**Authors:** Beata Joanna Mossakowska, Natalia Rusetska, Ryszard Konopinski, Paulina Kober, Maria Maksymowicz, Monika Pekul, Grzegorz Zieliński, Andrzej Styk, Jacek Kunicki, Mateusz Bujko

**Affiliations:** 1Department of Molecular and Translational Oncology, Maria Sklodowska-Curie National Research Institute of Oncology, 02-781 Warsaw, Poland; 2Department of Experimental Immunology, Maria Sklodowska-Curie National Research Institute of Oncology, 02-781 Warsaw, Poland; 3Department of Cancer Pathomorphology, Maria Sklodowska-Curie National Research Institute of Oncology, 02-781 Warsaw, Poland; 4Department of Neurosurgery, Military Institute of Medicine, 04-141 Warsaw, Poland; 5Department of Neurosurgery, Maria Sklodowska-Curie National Research Institute of Oncology, 02-781 Warsaw, Poland

**Keywords:** Cushing’s disease, corticotroph PitNET, pituitary, *USP8*, *USP48*, cell cycle, p27, palbociclib, roscovitine, flavopiridol

## Abstract

**Simple Summary:**

Patients with corticotroph pituitary tumors are basically treated with surgery, however, a relatively high rate of patients experience tumor recurrence. There are no clear standards of pharmacological adjuvant treatment of recurrent corticotroph tumors. Few approaches are considered including the strategy of targeting the regulation of cell cycle progression with inhibitors of cyclin-dependent kinases. Approximately 40% of patients have mutations in *USP8* or *USP48* genes that have a strong effect on tumor biology, including changes in the expression of cell cycle-related genes. Our goal was to assess whether deubiquitinase gene mutations could possibly serve as an indicator for treatment with inhibitors of the cell cycle. It could potentially allow for more rational and personalized treatment of the patients.

**Abstract:**

Protein deubiquitinases *USP8* and *USP48* are known driver genes in corticotroph pituitary neuroendocrine tumors (PitNETs). *USP8* mutations have pleiotropic effects that include notable changes in genes’ expression. Genes involved in cell cycle regulation were found differentially expressed in mutated and wild-type tumors. This study aimed to verify difference in the expression level of selected cell cycle-related genes and investigate their potential role in response to cell cycle inhibitors. Analysis of 70 corticotroph PitNETs showed that *USP8*-mutated tumors have lower *CDKN1B*, *CDK6*, *CCND2* and higher *CDC25A* expression. *USP48*-mutated tumors have lower *CDKN1B* and *CCND1* expression. A lower p27 protein level in mutated than in wild-type tumors was confirmed that may potentially influence the response to small molecule inhibitors targeting the cell cycle. We looked for the role of *USP8* mutations or a changed p27 level in the response to palbociclib, flavopiridol and roscovitine in vitro using murine corticotroph AtT-20/D16v-F2 cells. The cells were sensitive to each agent and treatment influenced the expression of genes involved in cell cycle regulation. Overexpression of mutated *Usp8* in the cells did not affect the expression of p27 nor the response to the inhibitors. Downregulating or upregulating p27 expression in AtT-20/D16v-F2 cells also did not affect treatment response.

## 1. Introduction

Pituitary neuroendocrine tumors (PitNETs) are frequently occurring intracranial neoplasms. They originate from various kinds of secretory pituitary cells. Tumors derived from corticotroph cells (corticotroph PitNETs) commonly cause ACTH-dependent Cushing’s disease [1]. However, some of them do not secrete the excessive amount of active ACTH hormone. These clinically non-functioning tumors are therefore classified as subclinical/silent corticotroph adenomas (SCAs) [1]. Functioning and silent corticotroph tumors have comparable molecular profiles [2,3].

The discovery of recurrent *USP8* and *USP48* mutations [4,5,6] allowed researchers to get some insight in the pathogenesis of corticotroph PitNETs [7]. These mutations are observed in approximately 40% of patients suffering both from Cushing’s disease and silent tumors. Both genes encode deubiquitinase enzymes involved in the regulation of protein degradation in proteasomes. Mutations in *USP8* and *USP48* are single nucleotide changes or small in-frame deletions in hot spot regions. *USP8* mutations impair interactions between *USP8* and 14-3-3 proteins which normally suppress deubiquitinase activity [8]. Therefore, mutations in *USP8* enhance its activity and result in decrease of proteasomal degradation of particular proteins and dysregulation of natural protein turnover [4,5]. Similarly, *USP48* mutations also increase the deubiquitinase activity [9]. Given that these two deubiquitinating enzymes have many substrates, impairment of their function may affect many proteins and processes. The pleiotropic effect of *USP8* mutations is clearly reflected by altered gene expression profile in *USP8*-mutated corticotroph pituitary tumors as compared to *USP8* wild-type (*USP8*wt) corticotroph PitNETs [2,3].

Previous reports on expression profiling in *USP8*-mutated and wild-type tumors showed difference in the expression of the large number of genes and, importantly, the most differentially expressed genes included those involved in the regulation of the cell cycle including *CDKN1B*, *CCND2*, *CDK6* and *CDC25A* [2,3].

Cell cycle proteins, especially cyclin-dependent kinases (CDKs) are considered to be novel therapy targets for treatment of Cushing’s disease (CD) patients [10]. The efficacy of the cell cycle inhibitor roscovitine (seliciclib) was observed in vivo in a mouse model of corticotroph tumors obtained by subcutaneous injection of the cells of corticotroph tumor cell line and in primary culture of corticotroph tumor cells [11]. Clinical trials with roscovitine were started in CD patients [10].

The activity of small molecule inhibitors of CDKs may be related to the expression status of particular genes involved in cell cycle regulation [12,13].

The aim of this study was to verify whether *USP8*-mutated and wild-type corticotroph PitNETs differ in the expression of particular genes and proteins involved in cell cycle regulation and whether these differences could play a role in response to cell cycle inhibitors.

## 2. Materials and Methods

### 2.1. Patients Characteristics

The study included 47 patients with CD and 23 patients suffering from SCA. All patients with CD had clear clinical signs and symptoms of hypercortisolism verified according to biochemical criteria, including increased urinary free cortisol (UFC) in three 24 h urine collections, disturbances of cortisol circadian rhythm, increased serum cortisol levels accompanied by increased or not suppressed plasma ACTH levels at 8.00, and a lack of suppression of serum cortisol levels to <1.8 µg/dL during an overnight dexamethasone suppression test (1 mg at midnight). The pituitary etiology of Cushing’s disease was confirmed based on the serum cortisol levels or UFC suppression <50% with a high-dose dexamethasone suppression test (2 mg q.i.d. for 48 h) or a positive result of a corticotropin-releasing hormone stimulation test (100 mg i.v.) and positive pituitary magnetic resonance imaging. Patients with SCA had no clinical or biochemical signs of hypercortisolism and showed normal levels of midnight cortisol and 24 h UFC. Of note, in this group of SCA patients, the proportion of invasive tumors was not as high as generally observed in subclinical corticotroph tumors [14]. Similarly, in our study’s SCA group, there is no higher female prevalence [14]. This indicates that group of SCA patients in this study is rather unrepresentative in terms of clinical characteristics.

ACTH levels were assessed using IRMA (ELSA-ACTH, CIS Bio International, Gif-sur-Yvette Cedex, France). The analytical sensitivity was 2 pg/mL (reference range: 10–60 pg/mL). Serum cortisol concentrations were determined by the Elecsys 2010 electrochemiluminescence immunoassay (Roche Diagnostics, Mannheim, Germany). Analytical sensitivity of the assay was 0.02 μg/dL (reference range: 6.2–19.4 μg/dL). UFC was determined after extraction (liquid/liquid with dichloromethane) by an electrochemiluminescence immunoassay (Elecsys 2010, Roche Diagnostics, Mannheim, Germany)—reference range: 4.3–176 μg/24 h.

All the tumor samples were ACTH-positive upon immunohistochemical staining against pituitary hormones (ACTH, GH, PRL, TSH, FSH, LH, α-subunit) and had characteristic ultrastructural features of corticotroph tumors, as determined with electron microscopy. Forty-one adenomas were positive only for ACTH, while 7 ACTH-positive tumors showed additional moderate/weak immunoreactivity for α-subunit. Increased proliferation assessed by Ki-67 index ≥3% was observed in similar proportion of CD and SCA patients—in 7 tumors causing CD and 5 SCAs.

Patients’ characteristics are presented in Table 1. The study was approved by the local Ethics Committee of Maria Sklodowska-Curie National Research Institute of Oncology, Warsaw, Poland. Each patient provided informed consent for the use of tissue samples for scientific purposes.

DNA and total RNA from FFPE samples was isolated using the RecoverAll™ Total Nucleic Acid Isolation Kit for FFPE (no. AM1975, Thermo Fisher Scientific, Waltham, MA, USA) and measured using a NanoDrop 2000 (Thermo Fisher Scientific). The samples were stored at −70 °C. *USP8* and *USP48* mutations were identified as previously reported [15].

### 2.2. Quantitative Real-Time RT-PCR

One microgram of tumor-derived total RNA was subjected to reverse transcription with a Transcriptor First Strand cDNA Synthesis Kit (no. 04897030001, Roche Diagnostics). qRT-PCR reaction was carried out in a 384-well format using a 7900HT Fast Real-Time PCR System (Applied Biosystems, Foster City, CA, USA) and Power SYBR Green PCR Master Mix (no. A25742, Thermo Fisher Scientific) in a volume of 5 μL, containing 2.25 pmol of each primer. The samples were amplified in triplicates. *GAPDH* was used as a reference gene. The Delta Ct method was used to calculate relative expression level. PCR primers’ sequences are presented in Appendix A. 

### 2.3. Immunohistochemistry

Immunohistochemical staining (IHC) was performed on 4 μm FFPE tissue sections using the Envision Detection System (no. K500711-2, DAKO, Glostrup, Denmark). Sections were deparaffinized with xylene and rehydrated in a series of ethanol solutions of decreasing concentration. Heat-induced epitope retrieval was carried out in a Target Retrieval Solution pH 9 (DAKO) in a 96 °C water bath, for 30 min. Cooled slides were treated with a blocker of endogenous peroxidase (DAKO) for 5 min and subsequently incubated with the primary antibodies as indicated in Table 2. The color reaction product was developed with 3,3′-diaminobenzidine tetrahydrochloride (DAKO) as a substrate, and nuclear contrast was achieved by hematoxylin counterstaining. Analysis of immunohistochemical reactivity was performed by calculating the H-score, which combines information on both reaction intensity (scored from 0 to 3) and the number of the cells with a given intensity. A previously reported formula was used for quantification [16]. Scoring results were analyzed as continuous variables. For each SCA sample, immunostaining against T-PIT was performed to verify the corticotroph nature of the tumors.

### 2.4. Cell Line Culture, Transfection and Cell Cycle Inhibition

AtT-20/D16v-F2 cells were purchased from ATCC collection and cultured in DMEM medium (no. 31966021, Gibco/Thermo Fisher Scientific, Waltham, MA, USA) supplemented with 10% FBS (no. 10270106, Gibco/Thermo Fisher Scientific) and 1% penicillin–streptomycin antibiotic solution (no. 15140122, Gibco/Thermo Fisher Scientific) according to supplier’s recommendations.

Three days before transfection 5 *×* 10^3^ cells were seeded per well into a 96-well plate in 100 μL culture medium. Afterwards, cells were transfected with 100 ng of plasmid, using 0.25% (*v/v*) lipofectamine 3000 (no. L3000008, Invitrogen, Carlsbad, CA, USA) or 0.45% (*v/v*) lipofectamine 2000 (no. 11668019, Invitrogen), according to the manufacturer’s instructions. Plasmids used for transfection are listed in Table 3 After 48 h, the medium was replaced with medium containing palbociclib, roscovitine or flavopiridol. Step-wise increasing concentrations of each inhibitor were used, as shown in Table 4. Cells were incubated for 24 h and then a cell viability assay or cell proliferation assay were performed. At least three independent experiments were conducted.

Protein overexpression was induced using a pcDNA3.1 expression plasmid. Coding sequences of *Usp8* (NM_001252580) and *Cdkn1b* (NM_009875) were PCR-amplified using cDNA from AtT-20/D16v-F2 cells as a template and cloned into multicloning site using the appropriate enzyme. A *Usp8* pP693R missense mutation that corresponds to human pP720R mutation was generated in PCR-amplified coding sequence with a QuikChange Lightning Site-Directed Mutagenesis Kit (no. #210519, Agilent, Santa Clara, CA, USA), following manufacturer’s protocol. 

Knockdown experiments were based on shRNA delivery in a plKO1gfp plasmid vector. shRNA was generated by annealing synthetic oligonucleotides and the subsequent ligation of double-stranded construct into a linearized plKO1gfp plasmid, according to the protocol. The sequences of PCR primers and oligonucleotides used for preparation of the plasmid construct are available in Appendix A.

### 2.5. Cell Viability Assay

Cell viability was measured with MTT reagent (no. 88417, Sigma-Aldrich, Burlington, MA, USA). Ten microliters of 5 mg/mL MTT stock solution was added to wells and cells were incubated for 4 h at 37 °C in cell culture incubator. Next, the medium was carefully removed and cells were flooded with 100 μL DMSO, and were mixed and incubated at 37 °C for 15 min. Absorbance was measured immediately at 540 nm on a Victor 3 microplate reader (Perkin Elmer, Wellesley, MA, USA).

### 2.6. Cell Proliferation Assay

Cell proliferation was measured with BrdU cell proliferation ELISA kit (no. 11647229001, Sigma-Aldrich), following the manufacturer’s protocol. The cells were incubated with BrdU reagent for 6 h at 37 °C in a cell culture incubator. Absorbance was measured at 450 nm and 690 nm using a Victor 3 microplate reader (Perkin Elmer).

### 2.7. Western Blotting 

Cells were lysed in ice cold RIPA buffer, incubated for 30 min in 4 °C and centrifuged at 14,400× *g* for 20 min at 4 °C. Samples were resolved using SDS-PAGE and electrotransferred to polyvinylidene fluoride membranes (PVDF) (Thermo Fisher Scientific). USP8 protein was detected with USP8 Rabbit polyclonal antibody (27791-1-AP, Proteintech, Rosemont, IL, USA), p27 protein was detected with Anti-p27 KIP1 antibody (ab32034, Abcam, Waltham, MA, USA). Reference proteins Lamin A/C and Glyceraldehyde-3-Phosphate Dehydrogenase were detected with Lamin A/C antibody (sc-20681, Santa Cruz Biotechnology, Dallas, TX, USA) and Anti-Glyceraldehyde-3-Phosphate Dehydrogenase (#MAB374, Sigma-Aldrich), respectively. Anti-rabbit antibody conjugated to HRP (#7074, Cell Signalling, Danvers, MA, USA) was used as secondary antibody. SuperSignal West Pico Chemiluminescent Substrate (no. 34087, Thermo Fisher Scientific) and the CCD digital imaging system Alliance Mini HD4 (UVItec Limited, Cambridge, United Kingdom) were used for visualization. Densitometry was analyzed with ImageJ software (National Institutes of Health, Bethesda, MD, USA). 

### 2.8. Statistical Analysis

A two-sided Mann–Whitney U-test was used for analysis of continuous variables. The Spearman correlation method was used for correlation analysis. A significance threshold of α = 0.05 was adopted. A two-sided Mann–Whitney U-test was also used for analysis on Western blot results. A one-way ANOVA with Bonferroni posttests was used for analysis of proliferation, survival and gene expression in AtT-20 cells and a 2-way ANOVA with Bonferroni posttests was used for analysis of proliferation and survival in AtT-20 transfected cells. Data were analyzed and visualized using GraphPad Prism 6.07 (GraphPad Software, San Diego, CA, USA).

## 3. Results

### 3.1. USP8 and USP48 Mutations in Corticotroph PitNETs

The sequences of *USP8* and *USP48* hot spot regions were analyzed with Sanger sequencing in 70 patients including 48 patients for whom the results of mutation screening were described previously [15]. *USP8* mutations were identified in 20/70 (28.57%) of the patients. They were missense variants of p.P720R (eight patients), p.P720Q (two patients), p.S718SP (two patients) and an in-frame deletion at position 719 (eight patients). *USP8* mutations were identified in four patients with subclinical CD, this result was reported previously [2]. *USP48* mutations were identified in five (7.14%) patients. All *USP48* mutations were p.M415I substitutions and were identified in CD patients. *USP8* and *USP48* mutations were mutually exclusive and all were heterozygous. 

### 3.2. The Expression of Genes Encoding for Cell Cycle Regulators in Corticotrph PitNETs Stratified According to USP8 and USP48 Status

The expression levels of four genes related to cell-cycle regulation were determined with qRT-PCR in tumor tissue from 70 patients with corticotroph tumors including patients with *USP8* mutations (n = 20), *USP48* mutations (n = 5) and wild-type patients (n = 42). The expression levels of *CDKN1B*, *CDK6*, *CCND2* and *CDC25A* were analyzed since these genes were previously found to be the most differentially expressed in corticotroph PitNETs with and without *USP8* mutations [2,3].

We observed significant differences between *USP8*-mutated and wild-type tumors in the expression of *CDKN1B*, *CDK6* and *CCND2* genes that showed significantly lower levels in mutated tumors (*p* < 0.0001, *p* = 0.0016, *p* < 0.0001, respectively), as well as for *CDC25A,* with a lower expression in the wild-type ones (*p* = 0.0005). Tumors with *USP48* mutations revealed a similar expression pattern of the analyzed genes as those with *USP8* variants. The levels of both *CDKN1B* and *CCND2* were significantly lower in *USP48*-mutated than in wild-type tumors (*p* = 0.0007 and *p* = 0.0005, respectively) while the difference in *CDK6* and *CDC25A* levels did not cross the significance threshold. Details are presented in Figure 1.

### 3.3. The Expression of Cell-Cycle Related Proteins in Corticotroph PitNETs

We determined the expression of the proteins encoded by *CDKN1B* (p27), *CDK6*, *CCND2* (cyclin D2) and *CDC25A* genes with immunohistochemical staining. Clear p27 nuclear immunoreactivity was observed. CDK6 showed mainly nuclear expression, however, weak cytoplasmic reactivity was also observed in most samples. Both nuclear and cytoplasmic cyclin D2 expression was observed, while CDC25A exhibited cytoplasmic reactivity. Immunoreactivity was quantified using the H-score formula, nuclear staining intensity and the percentage of positive cells were evaluated for determining expression of p27, CDK6 and cyclin D2, whereas cytoplasmic immunoreactivity was taken into account for the assessment of CDC25A expression. The results are presented in Figure 2.

We observed a significantly lower level of p27 in both *USP8*-mutated and *USP48*-mutated tumors as compared to wild-type patients (*p* < 0.0001 and *p* = 0.0036, respectively). A significant correlation between protein expression and gene expression was found in the entire set of tumor samples (Spearman correlation R = 0.536, *p* = 0.0078). The differences in the expression of other cell cycle genes were not observed at protein level. A slight but nonsignificant decrease in CDK6 expression was observed in *USP8*-mutated tumors. No significant difference was also observed in the expression of Cyclin D2 (encoded by *CCND2*) or CDC25A at protein levels. A correlation between the *CDKN1B* expression levels and P27 immunoreactivity score was observed (Spearman R = 0.536; *p* = 0.0003) while no significant correlation between gene and protein expression was found for *CDK6*, *CCND2* and *CDC25A*.

### 3.4. Treatment of AtT-20/D16v-F2 Cells with Cell Cycle Inhibitors Palbociclib, Roscovitine and Flavopiridol

AtT-20/D16v-F2 mouse corticotroph cells were treated with palbociclib, roscovitine or flavopiridol, which are small molecule inhibitors of cyclin-dependent kinases (CDKs) with a distinct spectrum of molecular targets (Figure 3A). Inhibitors of the cell cycle reduced proliferation and survival, with a dose-dependent effect (Figure 3B). High concentrations of flavopiridol (0.5, 1 µM) strongly affect proliferation, but have a lower influence on cell survival (Figure 3B).

We measured the expression level of selected cell cycle-related genes including *Cdkn1b*, *Cdkn1a* (both code for the cyclin-dependent kinase inhibitors, CDKIs, p27 and p21, respectively), *Ccnd1*, *Ccnd2* and *Ccnd33* (cyclin D genes), *Ccne1* and *Ccne2* (cyclin E genes), and *Cdk4*, *Cdk6* and *Cdc25a* in cells after treatment with each inhibitor. Treating the cells with palbociclib resulted in increased expression of *Cdkn1a* as well as cyclin D1 gene (in high inhibitor concentration). It also reduced the expression of genes encoding cyclin E and *Cdc25a*. Roscovitine treatment caused an increase in the expression of genes encoding both CDKIs and cyclins. No difference was observed in *Cdc25a* and *Cdk4* as compared to untreated cells. The use of flavopiridol increased the expression of CDKIs, cyclin E genes and *Cdc25a,* and decreased the expression of *Cdk4*.

The expression of *Ccnd2* and *Cdk6* in AtT-20/D16v-F2 cells was below qRT-PCR detection level. Very low expression of these genes in AtT-20 cells was confirmed using deposited RNAseq data (GSM3856431 dataset, Gene Expression Omnibus, not shown).

### 3.5. Treatment of AtT-20/D16v-F2 Cells Harboring Usp8 Mutation with Cell Cycle Inhibitors

AtT-20/D16v-F2 corticotroph cells were transfected with the plasmid vector pCDNA3.1, containing mutated *Usp8* (NM_001252580.1) and with wild-type *Usp8* in parallel. The overexpression of both mutated and wild-type *Usp8* variants was confirmed with Western blot (Figure 4A). Overexpression of mutated *Usp8* did not affect the cell proliferation rate as compared to cells with overexpression of wild-type *Usp8* (Figure 4C). 

The cells were also analyzed for the expression levels of *Cdkn1b*, *Cdk6*, *Ccnd2* and *Cdc25a*. We noticed that the presence of mutated *Usp8* did not affect the expression levels of *Cdkn1b* nor *Cdc25a* as observed in human CD patients. No difference between cells overexpressing mutated and wild-type *Usp8* was observed (Figure 4B). Expression of *Cdk6* and *Ccnd2* was below the qRT-PCR detection level both in cells with and without mutation. Unfortunately, we also did not notice either a significant difference in cell viability or cell proliferation between the AtT-20/D16v-F2 cells overexpressing wild-type and mutated *Usp8* that were treated with cell cycle inhibitors (Figure 4D,E).

### 3.6. Treatment of AtT-20/D16v-F2 Cells with Modified p27 Expression Level with Cell Cycle Inhibitors

Since a difference in p27 protein expression was found in patient-derived *USP8*-mutated and wild-type corticotroph tumors, we engineered the expression of *Cdkn1b* in AtT-20/D16v-F2 cells and looked for a response to cell cycle inhibitors. Knockdown of p27 in AtT-20 cells was achieved with shRNAs. Two shRNA hairpins against mouse *Cdkn1b* gene (Refseq NM_009875.4) were used: one complementary to 3′UTR (sh_p27.1) and the other complementary to the coding sequence (sh_p27.2). They showed a slightly different effect on p27 downregulation, with more effective silencing using the sh_p27.2 hairpin (Figure 5A). Knockdown of p27 resulted in the significant increase in proliferation rate in AtT-20/D16v-F2 cells treated with the sh_p27.2 hairpin (Figure 5B) but it did not influence the response of the cells to palbociclib, roscovitine or flavopiridol. No differences in cell viability/metabolic activity as well as no difference in cell proliferation rate were found between the cells with the reduced p27 level and control cells treated with scrambled shRNA (Figure 4C,D).

The increase in p27 expression in AtT-20/D16v-F2 cells was achieved with transfecting the cells with plasmid vector with cloned *Cdkn1b* coding sequence. Efficient overexpression of p27 was confirmed with Western blot (Figure 6A). Overexpression of p27 reduced the proliferation rate as compared to control cells (Figure 6B). No difference in the response to inhibitors of CDKs was found in the cells with overexpression of p27 and control cells. No difference in cell viability/metabolic activity or cell proliferation was observed (Figure 6D).

## 4. Discussion

Mutations in genes encoding deubiquitinating enzymes play a role in development of corticotroph tumors. In our cohort, *USP8* mutations were identified in 28.57% of patients, while *USP48* mutations were identified in 7.14%. These proportions are similar to those previously reported [7]. *USP48* mutations were found in tumors negative for *USP8* mutations and this mutual exclusion was also described previously [7]. Our study included 70 patients and the clinical relevance of *USP8* mutations was already investigated based on the data from 48 of these patients [15].

Hot-spot mutations in both deubiquitinases enhanced deubiquitinating activity and led to the dysregulation of proteasomal degradation of particular proteins and dysregulation of natural protein turnover [4,5,6]. Although the direct effect of mutations is at the level of protein regulation, the striking difference in gene expression pattern was also observed in *USP8*-mutated and *USP8*-wild-type corticotroph tumors [2,3]. These considerable differences in the molecular background of mutated and wild-type tumors indicate the possibility that *USP8* mutational status may serve as predictive factor for treatment. It was already suggested that it may serve as an indicator of treatment with somatostatin analogs (due to difference in SSTR5 expression) [3,17,18,19] and temozolomide (due to *MGMT* expression) [3]. An improved pasireotide response in mutated patients was reported recently [20].

The treatment of corticotroph PitNETs is challenging. Surgery is the basic method of therapy but approximately ~35% of patients require adjuvant treatment [21]. In CD, therapeutical approaches include targeting pituitary or adrenal gland activity with somatostatin/dopamine analogs or glucocorticoid receptor blockers, respectively, as well as inhibitors of cortisol synthesis [21]. However, due to invasive nature of corticotroph tumors and the high recurrence rate, classic oncological pharmacotherapy is also adopted for treating both CD and subclinical corticotroph PitNETs. Alkylating agents such as temozolomide or immunotherapy were used in these groups of patients [22,23]. Regulation of cell cycle progression represents another therapeutic approach, which was considered for the treatment of corticotroph PitNETs. A few small molecule inhibitors of the cell cycle such as palbociclib, roscovitine or flavopiridol have been already approved for treatment of various human tumors [24]. Two clinical trials on the effectiveness of roscovitine in treatment of the patients with corticotroph PitNETs have been launched [21]. Roscovitine was shown to inhibit ACTH secretion and tumor growth/proliferation in corticotroph cells and animal models [11,25].

Genes encoding important cell cycle regulators such as *CDKN1B* (encoding p27), *CCND2* (cyclin D2), *CDK6* (cyclin-dependent kinase 6) and *CDC25A* are among the most differentially expressed in a comparison of the transcriptomic profile of *USP8*-mutated and wild-type tumors [2,3]. Our gene expression assessment with qRT-PCR confirmed a lower expression of *CDKN1B*, *CCND2*, *CDK6* and a higher expression of *CDC25A* in *USP8*-mutated than in wild-type patients. Since *USP48* somatic variants are much less frequent than *USP8* changes, their role is less recognized. Our results indicate a similar pattern of the expression of selected cell cycle-related genes, especially *CDKN1B* and *CCND2,* in tumors with mutations in either deubiquitinating enzyme. The observed differences in the expression of four particular genes suggest that different elements of cell cycle regulation may be dysregulated in mutated and wild-type tumors. This expression pattern suggests cyclin D activation in wild-type tumors (high *CCND2* and *CDK6*) and cyclin E activation in tumors with *USP8*/*USP48* changes (low *CDKN1B*). 

Evaluation of the expression of proteins encoded by the selected genes in mutated and wild-type tumors verified a clear difference in p27 level only, while no significant difference was observed in cyclin D2, CDK6 and CDC25A. A general decrease of p27 expression in corticotroph tumors was observed previously [26] and it appears that the level of this protein is notably lower in tumors with *USP8* or *USP48* mutations than in wild-type tumors. This result strongly supports previous observations by Weigand I et al., who reported lower p27 immunoreactivity in tissue samples of *USP8*-mutated tumors [27], and this contradicts the result by Martins CS et al., who didn’t find such a difference [28]. Importantly, the role of p27 expression level in the response to CDKs inhibitors was already reported. Lowering the p27 level results in a better treatment response to palbociclib due to a decline in the activating effect of p27 on CDK4/cyclin D [12]. It also seems to improve the response to flavopiridol as observed in mice with Cdk4 mutations, indicating the role of p27 in response to flavopiridol. These mice developed pituitary adenomas and flavopiridol was significantly more effective in this model when p27 expression was decreased [29]. 

Our in vitro results on AtT-20 cells confirm that palbociclib, roscovitine and flavopiridol inhibit proliferation and viability. Treatment with inhibitors induces changes in the expression level of cell cycle-related genes indicating the role of these genes in the response to treatment. Differences in the gene expression in cells treated with particular inhibitors seem to reflect their different mechanisms of action. For example, palbociclib acts mainly on cyclin D-CDK4/6 complex in G1 phase of the cell cycle. Cyclin D triggers the expression of cyclin E and transition of cells into the S phase. AtT-20 cells treated with palbociclib showed an increased expression of cyclin D genes and decrease in cyclin E that probably results from impaired activation of the expression of cyclin E genes due to inactivation of CDK4. In turn, treatment with roscovitine caused increased expression of both cyclin D and cyclin E genes. Unlike palbociclib, roscovitine acts on CDK1/CDK2 in the S phase of the cell cycle so it does not downregulate cyclin E genes. Expression of *Cdkn1b* was increased in cells treated with roscovitine and flavopiridol, suggesting that this gene may play a role in the response to these inhibitors.

To evaluate the possible effect of *USP8* mutations in response to cell cycle inhibitors we investigated whether palbociclib, roscovitine or flavopiridol may have different effects in AtT-20 corticotroph cells with *Usp8* mutations or in cells with modified p27 expression. *Usp8*-mutated and wild-type variants were overexpressed in AtT-20 cells. Unfortunately, we did not observe any difference in p27 expression in cells with mutated and wild-type *Usp8* that would reflect different expressions in patients. We also did not find a difference in the response to treatment with CDK inhibitors. Downregulation of p27 is not a direct effect of mutations of deubiquitinating enzymes but an indirect consequence of change in EGFR–MAPK signaling in *USP8*-mutated corticotroph tumors, as concluded previously [5]. Therefore, we assume that this indirect effect was not achieved in the laboratory model and overexpressing mutated *Usp8* in AtT-20 cells do not mimic the effect of *USP8* mutation observed in patients. For evaluation of the role of p27 in treatment response, we directly manipulated this protein expression in AtT-20 cells and treated these cells with CDKs inhibitors. Unfortunately, neither increasing nor decreasing the p27 level affected the response to palbociclib, roscovitine or flavopiridol. Thus, our results of in vitro experiment do not confirm the hypothesis that p27 level determines the drug effect. In general, the studies on the treatment mechanism should not be based on a single cell line model and, commonly, the experiments on drug response are performed in multiple cell lines in parallel. This is not available when investigating corticotroph tumors, since AtT-20 is the only available authenticated cell line model of corticotroph tumors.

## 5. Conclusions

Mutations in genes encoding protein deubiquitinases occur in corticotroph PitNETs and mutated and wild-type tumors differ in the expression levels of genes related to cell cycle regulation. Protein expression of p27 in *USP8-* and *USP48*-mutated tumors is notably lower than in the tumors without mutations. Since p27 plays a role in the response to small molecule inhibitors of CDKs it could be hypothesized that wild-type patients and those with mutated deubiquitinases may exhibit different response to cell cycle-targeting treatment.

Our results of in vitro experiments with AtT-20 cells, which are the only available cell line model of corticotroph pituitary tumors, do not confirm this assumption. Treating these cells with palbociclib, roscovitine and flavopiridol reduced proliferation and viability. It also affected the expression of genes involved in cell cycle regulation but the response to treatment did not depend on p27 expression level.

## Figures and Tables

**Figure 1 cancers-14-05594-f001:**
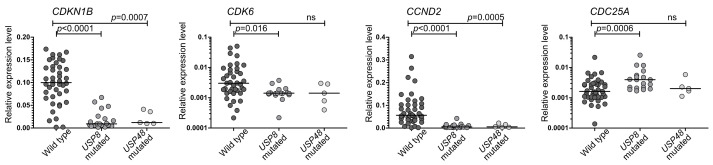
The expression of selected genes encoding cell cycle-related proteins in corticotroph PitNETs stratified according to deubiquitinase gene mutation status.

**Figure 2 cancers-14-05594-f002:**
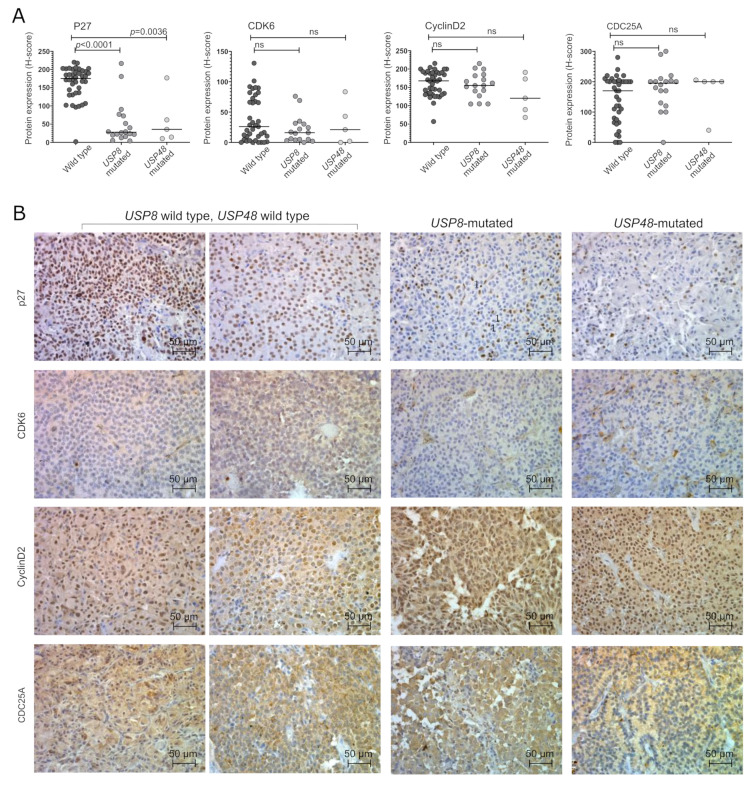
Expression of cell cycle-related proteins in corticotroph PitNETs stratified according to deubiquitinase gene mutations; (**A**). The results of the quantification of immunoreactivity; (**B**). The examples of staining results.

**Figure 3 cancers-14-05594-f003:**
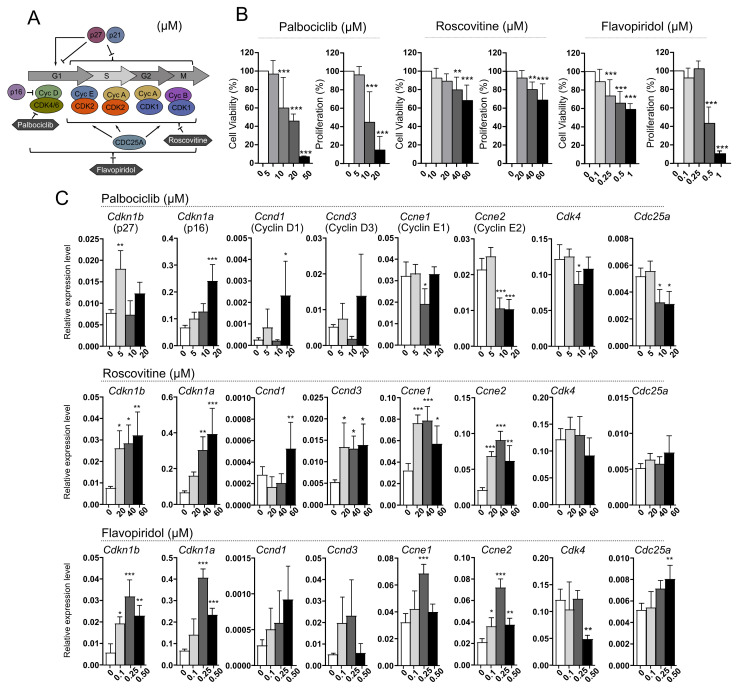
The effect of inhibitors of cyclin-dependent kinases (CDKs), palbociclib, roscovitine and flavopiridol, on AtT-20/D16v-F2 corticotroph cells. (**A**). Simplified scheme of cell cycle regulation indicating the specificity of each CDK inhibitor; (**B**). The effect of cell cycle inhibitors on viability and proliferation of AtT-20/D16v-F2 cells. Difference in viability and proliferation of treated cells was evaluated in comparison to untreated cells. ** indicates *p* < 0.001, *** indicates *p* < 0.0001; (**C**). The expression of genes encoding key cell cycle regulators in AtT-20/D16v-F2 cells treated with palbociclib, roscovitine and flavopiridol. Difference in the gene expression level in treated cells was evaluated in comparison to untreated cells. * indicates *p* < 0.05, ** indicates *p* < 0.001, *** indicates *p* < 0.0001.

**Figure 4 cancers-14-05594-f004:**
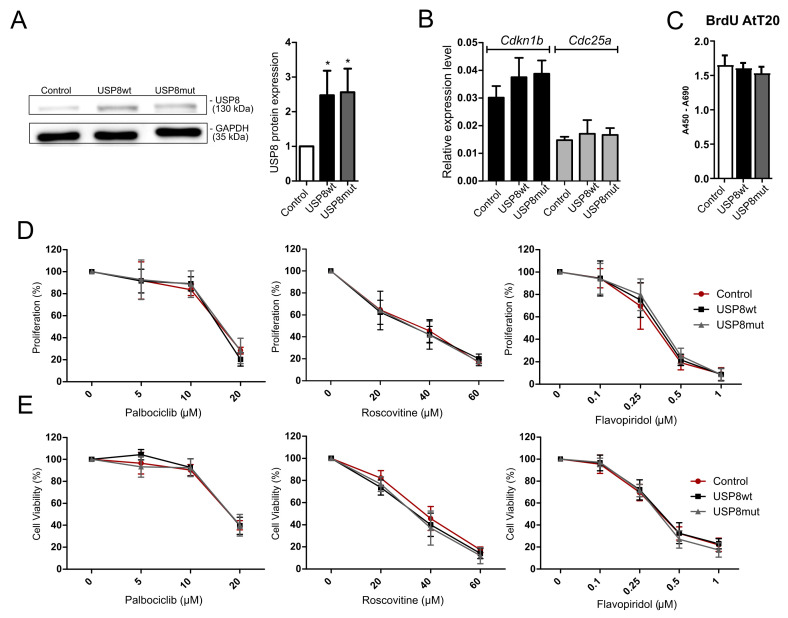
Treatment of AtT-20/D16v-F2 cells with overexpression of *USP8*-mutated and wild-type variant with cyclin-dependent kinase inhibitors (CDKIs); Control—empty vector, USP8wt—wild type USP8 overexpression; USP8mut—overexpression of mutated *USP8*. (**A**). Overexpression of mutated and wild-type Usp8 in AtT-20/D16v-F2 cells (Western blot membrane and densitometry results) * indicates *p* < 0.05; Original western blot figures can be found at Appendix A; (**B**). The expression levels of *Cdkn1b* and *Cdc25a* in AtT-20/D16v-F2 cells with overexpression of *Usp8*-mutated and wild-type variants. *Cdk6* and *Ccnd2* expression levels in AtT-20/D16v-F2 cells were not determined since they were below qRT-PCR detection.; (**C**). Proliferation of AtT-20/D16v-F2 cells with overexpression of *Usp8*-mutated and wild-type variants; (**D**). Proliferation rate of AtT-20/D16v-F2 cells with overexpression of *USP8*-mutated and wild-type variants treated with CDKIs: palbociclib, roscovitine and flavopiridol, measured with the BrdU assay; (**E**). The viability of AtT-20/D16v-F2 cells with overexpression of *Usp8*-mutated and wild-type variants treated with CDKIs: palbociclib, roscovitine and flavopiridol, measured with the MTT assay.

**Figure 5 cancers-14-05594-f005:**
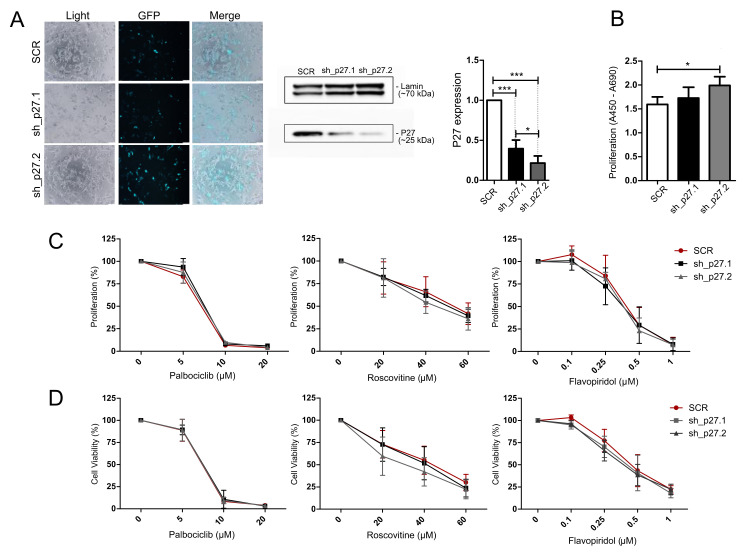
Treatment of AtT-20/D16v-F2 cells with p27 knockdown using cell cycle inhibitors. (**A**). Transfection efficiency and lowering of p27 expression with the shRNA hairpins sh_p27.1 (targeting 3′UTR) and sh_p27.2 (targeting coding sequence). Microscopic pictures were taken by the use of Olympus CKX53 with 10 × objective. p27 expression level as determined with Western blot and quantified with densitometry; * indicates *p* < 0.05, *** indicates *p* < 0.0001. (**B**). Change of proliferation rate in AtT-20/D16v-F2 cells with efficient p27 downregulation.; * indicates *p* < 0.05. (**C**). The effect of the cell cycle inhibitors palbociclib, roscovitine and flavopiridol on the proliferation rate of cells with downregulated p27 and control cells (scrambled shRNA, SCR) as measured with the BrdU assay. (**D**). The effect of cell cycle inhibitors on the viability of cells with downregulated p27 and control cells (SCR) as measured with the MTT assay.

**Figure 6 cancers-14-05594-f006:**
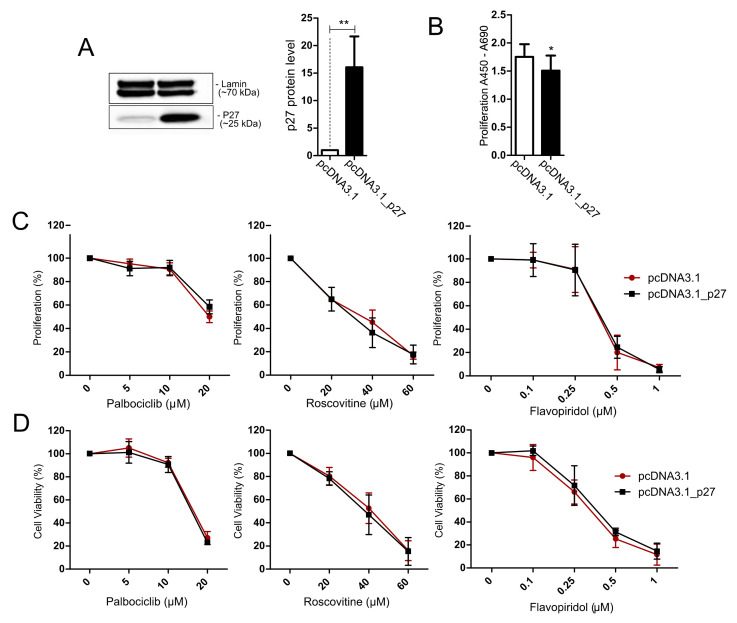
Treatment of AtT-20/D16v-F2 cells overexpressing p27 with cell cycle inhibitors. (**A**). Expression of p27 in AtT-20/D16v-F2 cells after transfection with expression plasmid vector with *Cdkn1b* coding sequence (pcDNA3.1_p27) and control cells (transfected with empty vector pcDNA3.1) as determined with Western blot and quantified with densitometry; ** indicates *p* < 0.001; (**B**). Proliferation rate in AtT-20/D16v-F2 cells with p27 overexpression and control cells; * indicates *p* < 0.05; (**C**). The effect of the cell cycle inhibitors palbociclib, roscovitine and flavopiridol on proliferation rate of cells overexpressing p27 and control cells as measured with the BrdU assay. (**D**). The effect of cell cycle inhibitors on viability of cells with p27 overexpression and control cells, measured with the MTT assay.

**Table 1 cancers-14-05594-t001:** Summary of clinical features of patients with Cushing’s disease and silent corticotroph tumors.

Clinical Feature	Cushing’s Disease	Silent Corticotroph Tumors
Number of patients	n = 47	n = 23
Sex (ratio females/males)	38/9	12/11
Age at surgery (years; median (range)	44 (20–78)	54 (23–77)
Cortisol 08:00 h (µg/dL; median (range)	25.65 (11.54–49.7)	16.6 (6.8–50.8)
Cortisol 24:00 h (µg/dL; median (range)	19.35 (6.83–36.5)	0.9 (0.3–14.1)
ACTH 08:00 h (pg/dL; median (range)	76.05 (35.6–207)	47.7 (14.7–96.1)
UFC (μg/24 h; median (range)	490 (163.9–961.8)	94.7 (13.7–139)
Tumor volume (mm^3^; median (range)	873.3 (13.5–12,000)	3465 (900–11,088)
Invasive tumor growth (Knosp grade 0, 1, 2/3, 4)	33/14	19/4
Pathomorphology (sparsely/densely granulated)	14/33	14/9

**Table 2 cancers-14-05594-t002:** The list of antibodies used for immunohistochemistry and incubation conditions.

Antibody	Incubation Conditions
CDC25A (MA513794; Invitrogen)	Dilution 1:100; overnight incubation in 4 °C
CDK6 (PA5-87490; Invitrogen)	Dilution 1:100; 1 h
P27 (PA5-32530; Invitrogen)	Dilution 1:100; 10 min
Cycl D2 (PA5-95922; Invitrogen)	Dilution 1:100; 1 h
T-PIT (AMAB91409; Sigma-Aldrich)	Dilution 1:2000; 30 min

**Table 3 cancers-14-05594-t003:** Plasmids used for transfections.

Experiment	Plasmids
Overexpression of *USP8* and *USP8* with P720R mutation	pcDNA3.1 pcDNA3.1_USP8 pcDNA3.1_MMUT6
Knockdown of p27 expression	plKO1gfp_SCR plKO1gfp_p27.2 plKO1gfp_p27.3
Overexpression of p27	pcDNA3.1 pcDNA3.1_p27

**Table 4 cancers-14-05594-t004:** Cell cycle inhibitors.

Inhibitor	Target	Concentrations (μM)	Supplier
Palbociclib	CDK4, CDK6	5, 10, 20	Sigma-Aldrich (no. PZ0199)
Roscovitine	CDK1, CDK2	20, 40, 60	Sigma-Aldrich (no. 557364)
Flavopiridol	Multiple CDKs, CDC25	0.1, 0.25, 0.5	Sigma-Aldrich (no. F3055)

## Data Availability

The data presented in this study are available on request from the corresponding author.

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
