# Peer review of "The Expression of Cell Cycle-Related Genes in *USP8*-Mutated Corticotroph Neuroendocrine Pituitary Tumors and Their Possible Role in Cell Cycle-Targeting Treatment"

_cancers, 2022, doi:10.3390/cancers14225594_

Round 1

Reviewer 1 Report

The authors provided a possible role of deubiquitinase gene mutations in therapy targeting cell cycle in Corticotroph PitNETs. About 70 corticotroph pituitary neuroendocrine tumors (PitNETs) were considered for the experiment including USP8 mutation (n=20) , USP48 mutation (n= 5).  The expression levels of 4 genes related to cell-cycle regulation was determined using qRT-PCR of patient sample and compared against normal patient. 

qPCR results showed USP8 mutated tumors have lower CDKN1B, CDK6, CCND2 and higher CDC25A expression. USP48 mutated tumors mostly follow the same results. However, author didn’t observe any significant correlation between gene and protein expression of CDK6, CCND2  and CDC25A except for CDKN1B. There was no further analysis on the missing link.

Based on the above result author analyzed the effect of CDKs inhibitors palbociclib, roscovitine and  flavopiridol on AtT-20/D16v-F2 cells that resulted in reduced cell survival and proliferation in a dose dependent manner and increased expression of Cdkn1a as well as cyclin D1 gene

Palbociclib -- increased expression of Cdkn1a as well as cyclin D1 gene

                        reduced expression of genes encoding cyclin E and Cdc25a

Roscovitine --increase in the expression of genes encoding both CDKIs and cyclins

                        No difference was observed in Cdc25a and Cdk4 as compared to untreated cells

Favopiridol --increased the expression of CDKIs and cyclin E  and Cdc25a genes

                      decreased expression of Cdk4.

However, no correlation between protein and gene expression was observed (or reported).

To mimic patient sample they over expressed wildtype and mutated USP8 in AtT-20/D16v-F2 corticotroph cells (Fig 4). However, CDKs inhibitors had no impact on expression of Cdkn1 and cdc25a  among control and USP8-WT and mutant cells. 

Author nicely designed reverse experiment where they Knocked out  p27 (Fig 5) using shRNA and however, observed no difference in cell viability and proliferation between control and KO cells in presence of inhibitors. Interestingly overexpression of p27 (Fig 6) also had no different on cell viability and proliferation among control and KO cells in presence of CDKs inhibitors. 

There are well studied reports on  impact of CDKs inhibitors on cell cycle  and have clinical trial on patients with corticotroph PitNETs.  

In this study author tried to evaluate the patient data in invitro model system however, due to limitation of the model it was difficult to mimic the patient study. Although author designed experiments nicely to understand in dept the results and has honest explanation on negative results.

Specific comments:

1.     Change ‘ultrastructural’ in line 109 to microscopic.

2.     Figure 4: please include data about CCND2 and CDC25A expression.

3.     Section 3.6: please fix figure number. It should be Figure 5 (not 4). 

4.     Line 324: please change ‘decrease’ to ‘increase’.

5.     Lines 459-460: please fix typographical errors and complete the sentence. 

Reviewer 2 Report

The authors present a possible role of deubiquitinase gene mutations in therapy targeting cell cycle in corticotroph neuroendocrine pituitary tumors. They looked for the role of USP8 mutations or changed p27 level in the response to plabocyclib, flavopirydol and roscoivitine in vitro using murine corticotroph AtT-20/D16v-F2 cells. The authors should pay attention to the following detail:

The study included 47 patients with CD and 23 patients suffering from SCA. Patients with SCA had no clinical or biochemical signs of hypercortisolism and showed normal levels of midnight cortisol and 24h UFC, and the tumor samples were ACTH-positive upon immunohistochemical staining, without T-PIT staining. Patients characteristics are presented in Table 1. 23 SCA include 12 females and 11 males, and 19 Knosp grade 0, 1, 2, only 4 Knosp grade 3, 4. However, These results are not consistent with the literature reports. The 2017 World Health Organization classification of endocrine tumors defines pituitary adenomas based on their cell lineages. T-PIT can serve as a complimentary tool for further identification of SCA. The prevalence of SCA increased with the introduction of T-PIT. SCA showed increased invasion and aggressive SCA tended to affect female patients. How does the authors explain this inconsistency?

Reviewer 3 Report

In the current manuscript, Mossakowska et al. aimed to determine whether USP8 gene mutation govern response to cell cycle inhibitors in corticotroph pituitary tumors. First, they verified that cell cycle genes and proteins are deregulated in patient tumors with USP8 or USP48 gene mutations. Next, they assessed the importance of USP8 in response to cell cycle inhibitors by expressing mutant USP8 in  murine AtT-20 cells, a model for studying corticotroph pituitary cancer.  They observed that neither USP8 mutation nor downregulation of p27 (another marker in corticotroph tumors downstream of USP8) changed response to cell cycle inhibitors. Overall, the manuscript is well written however the key novelty of the manuscript i.e. using mutant USP8 to study response to cell cycle inhibitors has several limitations and need to be addressed before full acceptance.

Major comments:

1.       The title of the manuscript is misleading.

2.       Need validation of the USP8 mutant expressing AtT-20 cells: Authors need to show that the mutant USP8 is active in these cells by using a downstream assay for USP8 activity.

3.       Key finding is based on a weak assessment: MTT cell viability assay is a metabolic assay and should not be the ultimate assay in characterizing response to cytotoxic agents as there are other ways to assess response. For example: colony formation assay, which measures reproductive capability of the cell and sometimes can be more relevant in studying response to cytotoxic agents. Here is a recent reference paper just showing this, PMID: 35178190. Authors need to perform additional assay for cell survival and response to cell cycle inhibitors beside MTT assay.

4.       Need to use other systems: I agree with authors that AtT-20 is the only cell line (that too murine) available to study corticotroph pituitary tumors. However, authors can at the least study whether USP8 mutation affect response to cell cycle inhibitors in normal human cells. They can express the mutant USP8 in couple of different normal human cell lines and determine if there is any effect of USP8 mutation on response. Other in vivo alternatives are PTTG zebra fish model and mouse model.

Minor comments:

1.       Figure 5 is annoted as figure 4. Need to change that in figure legend and in the text.

2.       Please provide catalog numbers for reagents and kits

3.       Spelling mistakes: CDK1b (line 36), plabocyclib (line 38), found (line 221) among others.
